# Effect of Wild Strawberry Tree and Hawthorn Extracts Fortification on Functional, Physicochemical, Microbiological, and Sensory Properties of Yogurt

**DOI:** 10.3390/foods12183332

**Published:** 2023-09-05

**Authors:** Teresa Herrera, Maite Iriondo-DeHond, Ana Ramos Sanz, Ana Isabel Bautista, Eugenio Miguel

**Affiliations:** Área de Investigación Agroalimentaria, Instituto Madrileño de Investigación y Desarrollo Rural, Agrario y Alimentario (IMIDRA), 28805 Alcalá de Henares, Spain

**Keywords:** wild fruits extracts, yogurt, phenolic compound, inhibition digestive enzymes, sensory analysis

## Abstract

The composition analyses and health-promoting properties (antioxidant capacity, antidiabetic, and antihypertensive properties) of wild fruit extracts and the effect of the incorporation of strawberry tree (STE) and hawthorn (HTE) extracts on the physicochemical, instrumental textural, microbiological, and sensory parameters of yogurts were evaluated. The incorporation of wild fruit extracts in yogurt increased antioxidant and antidiabetic properties (inhibition of digestive α-amylase, α-glucosidase, and lipase enzymatic activities) compared to the control, without decreasing their sensory quality or acceptance by consumers. The hawthorn yogurt (YHTE) showed the highest total phenolic content (TPC) and antioxidant capacity (ABTS and ORAC methods). Yogurts containing wild fruit extracts and dietary fiber achieved high overall acceptance scores (6.16–7.04) and showed stable physicochemical, textural, and microbiological properties. Therefore, the use of wild fruit extracts and inulin-type fructans as ingredients in yogurt manufacture stands as a first step towards the development of non-added sugar dairy foods for sustainable health.

## 1. Introduction

The consumption of wild edible plants has been an important resource for different places and times in human cultures and could be used for different objectives such as food, medicine, the production of materials, or magic rituals. Most of them grown in the Mediterranean region (including almost all of South Europe, North Africa, and West Asia) are traditionally consumed [1,2,3,4]. They may have a greater amount of nutrients (such as vitamins and minerals) and bioactive compounds (such as carotenoids and phenolic compounds) than cultivated species [5]. Some wild fruits are good sources of vitamins C, E, and provitamin A; for example, the fruits of *Arbutus unedo* L. are remarkable for being a source of vitamin C, with levels of around 100 mg/100 g. Regarding phenolic compounds, mainly phenolic acids, flavonols, and anthocyanins, a higher content was described in Mediterranean wild fruits than in blueberries and blackberries [4]. Wild edible plants continue to be gathered in Europe, which may be applied by the food industry for food innovation purposes. Exotic or unusual foods can provide different colors and flavors, as well as different bioactive compounds or a higher concentration of the same bioactive compounds [6].

The strawberry tree (*Arbutus unedo* L., Ericaceae) is one of the most frequent wild fruit species in the Mediterranean region. Its raw fruits are usually consumed directly. The interest in the health benefits of strawberry tree fruits or their fruit extracts to add as ingredients into yogurts, pie and pastry fillings, cereals, or meat products has been recently reported [7,8]. Strawberry tree fruits could also be employed as a food colorant, taking into account their content of β-carotene and anthocyanins [7], and can be used as a source of nutrients and bioactive compounds such as vitamin C, dietary fiber, and phenolic compounds (gallic acid and anthocyanins as cyanidin 3-glucoside) [6,9,10]. These compounds are associated with antioxidant properties with health implications.

Hawthorn (*Crataegus monogyna*, Rosaceae) is the oldest known medicinal plant in European medicine. Many studies have shown that hawthorn wild fruit possesses potent antioxidant and free radical scavenging activities, attributable to the presence of different bioactive compounds, among which are polyphenols such as epicatechin, hyperoside, and chlorogenic acid [11]. These compounds present pharmacological effects such as neuroprotective, hepatoprotective, cardioprotective, and nephroprotective properties, among others [12]. The interest in using hawthorn in food applications such as yogurts and tonic wines has increased over the last few years [13,14].

Among fermented dairy products, yogurt is the most popular. It is considered a healthy food because of its high digestibility and bioavailability of nutrients [15]. In the last few years, the decrease in the consumption of dairy products has had a negative impact on this sector. In this sense, the dairy industry is looking to establish new innovative strategies for the development of new products to continue promoting this sector. The new trends towards the consumption of natural products make the incorporation of wild fruit extracts into dairy products a strategy to bring new consumers closer to dairy products. Consumers are more concerned about the nutritional value of the food they eat. Consequently, they look for foods that include natural products versus synthetic chemical compounds [16]. Plant extracts can provide beneficial health effects when they are incorporated into foods, mainly as antioxidants related to the prevention of chronic non-communicable diseases (diabetes, metabolic syndrome, or hypertension). In addition to these health benefits, the sensory properties (appearance, texture, and flavor) of the yogurt are important factors in its consumer acceptability. Ingredients such as milk protein, prebiotics, and herbs contribute to the nutritional variations and/or technical applications of yogurts [17].

The World Health Organization recommends reducing the sugar content of processed foods. Furthermore, yogurt is one of the commercial products with the highest amount of added sugar, producing an increase in the caloric value. To replace the sugar content in dairy products, a strategy may be the incorporation of prebiotics like inulin-type fructans, such as inulin and fructo-oligosaccharides (FOS). Inulin has been reported to improve dairy products’ texture, whereas FOS has been previously applied for its sweetening properties [17]. Likewise, incorporation of foods such as natural sources of bioactive compounds, mainly polyphenols, is another strategy reported for the contribution of health-promoting ingredients in the human diet. For this reason, the search for antioxidant compounds from natural sources to develop new yogurts is of great interest and shows a novel approach for these types of dairy products [18,19]. Regarding extracts of many plants, herbs, and fruits rich in bioactive compounds, they are frequently used as ingredients in dairy products for better nutritional and functional improvement.

This study proposed the development and characterization of aqueous extracts of strawberry tree and hawthorn for their introduction in the development of polyphenol-enriched yogurts. Therefore, this study aims to: (a) characterize the chemical composition, microbiological analysis, antioxidant properties, and inhibitory effect against digestive enzymes of wild fruit extracts; (b) characterize the chemical composition, microbiological analysis, antioxidant properties, and inhibitory effect against digestive enzymes of yogurts with selected wild fruit extracts; and (c) evaluate the consumer sensory acceptance and purchase intent of these yogurts elaborated with wild fruit extracts and also dietary fiber. In order to obtain the wild fruit extract for the development of new sustainable and health-promoting yogurts taking into account their biological and sensory properties, different extraction conditions were previously tested to improve the content of phenolic compounds, the antioxidant capacity, the possible beneficial effects for the prevention of the prevalence of chronic non-communicable disorders, and the sensory acceptance of the yogurts in which the extracts are included.

## 2. Materials and Methods

### 2.1. Raw Materials and Extracts

Wild fruits from two different species were used: strawberry tree (*Arbutus unedo* L.) (water content: 45%) and hawthorn (*Crataegus monogyna* L.) (water content: 78%). Both wild fruits were collected at Finca El Encín, Alcala de Henares (Madrid, Spain). Wild fruits were frozen to −40 °C for at least 48 h in glass bottles and lyophilized (Telstar LyoQuest-85 PLUS, Terrassa, Spain) at 0.015 mBar at −80 °C for 72 h. The freeze-dried samples were ground to obtain a homogeneous fine powder (<500 µm).

Wild fruit extracts were prepared by making an aqueous extraction with distilled water at 60 °C and stirring for 1 h (Appendix A). Conditions of extraction were selected by a preliminary test in our laboratory, according to the literature [14,20]. We have selected these extracts because, in general, they showed better properties for both types of wild fruits, a higher content of phenolic compounds, better antioxidant capacity, and greater inhibition capacity against the different enzymes studied in comparison to other extracts that we tested previously in our laboratory (Appendix A). Strawberry tree extract (STE) and hawthorn extract (HE) were stored at −20 °C until further analysis. Extracts were performed in triplicate.

### 2.2. Yogurt Preparations

Five set-type yogurt formulations were prepared. For the different yogurts, UHT cow milk (3.6% fat, 3% protein, and 4.8% carbohydrates) was put in a vat to inoculate the starter culture, YO-MIX 885 (Danisco DuPont, Brabrand, Denmark), composed of *Streptococcus thermophilus* and *Lactobacillus delbrueckii* subsp. *bulgaricus*. Also, inulin and fructooligossacharides (FOS) were both added at 4 g/100 mL (Orafti^®^GR, Beneo, Leuven, Belgium). Wild fruit extracts from strawberry tree and hawthorn were added to the milk at concentrations of 8 mg/mL extract and 12 mg/mL extract, yielding the following formulations: 8 mg/mL strawberry tree extract (YSTE-8), 12 mg/mL strawberry tree extract (YSTE-12), 8 mg/mL hawthorn extract (YHE-8), 12 mg/mL hawthorn extract (YHE-12), and a control without fruit extracts (YC). The same volume of distilled water that was used to incorporate the aqueous extracts of strawberry tree and hawthorn was added to the control yogurt. The final concentration that we added to elaborate yogurts was previously evaluated in our laboratory for the incorporation of winery byproduct extracts in yogurts [21]. In addition, the final concentration employed was the maximum concentration allowed to homogenize with the mixture of ingredients for the elaboration of the different yogurts. Strawberry tree yogurts showed a certain pink color, and hawthorn yogurts showed a certain orange color. After all ingredients were incorporated, this mixture was shaken and separated into pots of 20 g (Appendix A). Individual pots were incubated at 48 °C for 5 h. Yogurt manufacturing was performed in triplicate in three independent sessions to assure reproducibility of results. Samples were subsequently stored at 4 °C.

Yogurt samples for composition analyses and health-promoting properties assays were prepared by diluting each sample (2.5 g) in distilled water (10 mL); incubated in a water bath for 10 min (45 °C); and centrifuged to remove precipitated proteins. The supernatant was recovered, filtered with 0.45 µm nylon filters (Symta, Madrid, Spain), and stored at −20 °C. Yogurt samples for technological properties were not treated as described previously but were taken and tested directly.

### 2.3. Composition Analyses

The composition analyses of wild fruit extracts and yogurts containing wild fruit extracts included the determination of soluble proteins, reducing sugars, lactose, total phenolic content (TPC), and the identification and quantification of phenolic compounds.

The determination of soluble proteins was performed according to the Bradford method, using the micro-method format to determine protein concentration [22]. All measurements were performed in triplicate. Bovine serum albumin (BSA) was used as a standard. Results were expressed as mg BSA/g extract.

Reducing sugars were measured according to Adney and Baker using the DNS reagent method [23]. All measurements were performed in triplicate. Results were expressed as g glucose/100 g extract.

The determination of lactose content in yogurts was carried out using the CDR FOODLAB^®^ (CDR s.r.l, Florence, Italy) photometric analyzer according to the manufacturer’s instructions. All measurements were performed in triplicate. Results were expressed as g lactose/100 g extract.

The analysis of TPC in extract and yogurt samples was carried out following the Folin–Ciocalteu method [24]. A gallic acid (GA) calibration curve (0.01–1 mg/mL) was used for quantification, and results were expressed as mg equivalents of GA (GAE)/g sample. Measurements were performed in triplicate.

The identification and quantification of phenolic compounds were carried out by HPLC-MS. The identification of phenolic compounds was conducted in an Agilent 1200 HPLC-MS system using pure commercial standards or by comparison with mass spectra from the literature. Quantification of total phenolic compounds was performed on an Agilent G6530A Accurate Mass Q-TOF LC-MS (Agilent Technologies Inc., Santa Clara, CA, USA) equipped with a ZORBAX Eclipse XDB-C18 column (150 mm × 4.6 mm, 5 μm particle size). The injection volume was 20 μL. Phenolic compounds were monitored at 280 nm. Chromatographic separation was achieved at 40 °C using 0.1% formic acid in water (Phase A) and 0.1% formic acid in acetonitrile (Phase B) at a flow rate of 1 mL/min as follows: 0 min: 5% B; 20 min: 15% B; 30 min: 30% B; 35 min: 50% B; 37 min: 5% B; 40 min: 5% B. The mass spectrometer was acquired in ESI mode using JetStream technology. Identification was in negative polarity. The quantification of phenolic compounds was performed using calibration curves obtained from commercial standards whenever possible: catechin was used for flavonoids, chlorogenic acid was used for phenolic acids, and cyanidin glucoside was used for anthocyanins. Analyses were carried out by the Analysis Service Unit of the Institute of Food Science, Technology, and Nutrition (ICTAN, CSIC, Madrid, Spain).

### 2.4. Health-Promoting Properties

We analyzed the antioxidant capacity and antidiabetic and antihypertensive properties of wild fruit extracts and yogurts containing wild fruit extracts.

The overall antioxidant capacity of wild fruit extracts and their yogurt samples was analyzed using the following methods:2,2′-azino-bis (3-ethylbenzothiazoline-6-sulphonic acid) (ABTS) assay: The trapping capacity of cationic free radicals was evaluated using the method of radical ABTS•+ bleaching [25,26]. Aqueous solutions of Trolox (0.02–1 mM) were used for calibration. Absorbance was measured at 734 nm in an automated plate reader (BMG LABTECH, Ortenberg, Germany). All measurements were performed in triplicate, and results were expressed as µmol TE/g sample;Oxygen Radical Absorbance Capacity (ORAC) assay: The ORAC assay was applied according to the method using fluorescein as a fluorescence probe [27]. The procedure was carried out using an automated plate reader (BMG LABTECH, Ortenberg, Germany) equipped with a fluorescence detector set at excitation and emission wavelengths of 485 nm and 530 nm, respectively. Readings were taken every minute for 90 min at 37 °C. All measurements were performed in triplicate, and results were expressed as µmol TE/g sample.

The antidiabetic properties of wild fruit extracts and yogurt samples were analyzed by digestive enzyme inhibition assays:α-amylase inhibition assay: The inhibitory activity of extracts and yogurts against α-amylase was measured using starch as substrate for the Caraway-Somogyi iodine/potassium iodide method [28,29] with slight modifications adapted to microplate [30]. Results were expressed as percentages of α-amylase inhibition. All measurements were performed in triplicate. Acarbose was used as a positive control;α-glucosidase inhibition assay: The α-glucosidase inhibitory activity of wild fruit extracts and yogurts was analyzed [31,32,33]. Acarbose was used as a positive control (standard inhibitor). Results were expressed as a percentage of α-glucosidase inhibition. All measurements were performed in triplicate;Lipase inhibition assay: The inhibitory activity of extracts and yogurt samples against pancreatic lipase was measured by using 4-methylumbelliferyl oleate (4-MUO) as substrate [34,35]. Orlistat was used as a positive control. Results were expressed as a percentage of lipase inhibition. All measurements were performed in triplicate.

The antihypertensive properties of wild fruit extracts and yogurt samples were analyzed by the Angiotensin Converting Enzyme (ACE) inhibition assay. The inhibitory activity of ACE was measured by using N-[3-(2-Furyl)acryloyl]-Phe-Gly-Gly (FAPGG) as a substrate [36,37]. Results were expressed as a percentage of ACE inhibition.

### 2.5. Technological Properties

The technological characterization (physicochemical and microbiological parameters) of wild fruit extracts was used to evaluate their potential application as food ingredients. In yogurts, technological properties were used as an indicator of their general quality.

In relation to the microbiological quality of STE and HE, the following microbial genera were analyzed: (i) total mesophilic aerobic bacteria (Plate Count Agar (PCA)); (ii) molds and yeasts (Potato Dextrose Agar (PDA)); (iii) enterobacteria (Violet Red Bile Agar with Glucose (VRBG)); (iv) coliforms (Violet Red Bile Agar with Lactose (VRBL)); and (v) streptococci (KF Streptococcal Agar (KF)). Different incubation conditions were set for each analysis: (i) 32 °C for 72 h; (ii) 25 °C for 6 days; and (iii, iv, and v) 37 °C for 72 h. All assays were performed in sterile conditions and were carried out in triplicate. Results were expressed as log CFU/g extract.

Yogurt physicochemical characterization included pH, moisture, instrumental texture, apparent viscosity, and syneresis analyses. A pH meter (Hanna Instruments HI5521) was used to measure the pH values. Moisture content was determined as described in AOAC-925.10. Yogurt syneresis was calculated by centrifugation [38]. Results were expressed in percentages.

Textural parameters were measured using texture profile analysis (TPA) by means of a TA.XTplus Texture Analyzer (Stable Micro Systems, Godalming, UK). A back-extrusion test was performed while using a cylindrical stainless-steel probe (35 mm diameter). Yogurts for texture analysis were made directly into cylindrical containers (50 mm in diameter and 50 mm in height) so that their solid structure would be kept intact prior to the texture analysis. The probe penetrated the sample to a depth of 10 mm at 1 mm/s. The TPA instrument measured different parameters such as firmness (N) and consistency (Ns) that were calculated from the deformation curves using the Exponent E32 software version 4.0.9.0 (Stable Micro Systems, Godalming, UK). Measurements were performed in triplicate.

Yogurts for apparent viscosity were made directly into cylindrical containers (30 mm in diameter and 115 mm in height) and measured at 5 °C using an Anton Paar viscometer (Viscosímetro Rotacional, ViscoQC 100) equipped with spindle L4 and mixed for 90 s at 30 rpm. The apparent viscosity was measured in triplicate. Results were expressed as Pa s.

Bacterial counts of *Streptococcus thermophilus* and *Lactobacillus delbrueckii* ssp. *bulgaricus* were carried out in triplicate following the colony count technique [21]. L. *delbrueckii*, ssp. *bulgaricus* colonies were counted on Man Rogosa Sharpe (MRS) agar (Pronadisa) after aerobic incubation at 37 °C for 72 h. *S. thermophilus* colonies were counted on M17 agar (Pronadisa) after aerobic incubation at 37 °C for 48 h. Results were expressed as log CFU/mL of yogurt.

### 2.6. Hedonic and Sensory Analyses

Consumers (*n* = 104, 40 males and 64 females, age range from 18 to 64 years) were recruited at the Instituto Madrileño de Investigación y Desarrollo Rural, Agrario y Alimentario (IMIDRA) (Madrid, Spain). The participation of the consumers was voluntary, and no monetary compensation was given. Sensory evaluation was performed in different sessions. Yogurt samples were prepared for each session independently and consumed by participants after two or three days of shelf life.

Consumers conducted a hedonic test to rate the overall acceptance, odor, flavor, and texture of five samples: strawberry tree, hawthorn, and control yogurts. Yogurt samples (30 mL) were offered at 7 °C in individual glass containers coded with a three-digit number. Samples were served in blind conditions and in a completely randomized order. Samples were rated using interval linear graphical scales 10 cm long, scoring 1 (lowest)–10 (highest), and the average of the panelists’ scores was calculated [39]. The linear graphic scale provides continuous data limited by the precision of the measuring instrument used to reflect the results, which approximates a normal distribution and generates continuous data. This increases the possibility of meeting the requirements for parametric evaluations [40].

### 2.7. Statistical Analyses

Statistical analyses for extracts were compared by the paired Student’s *t*-test. Statistical analyses for composition, health-promoting properties, technological properties, and hedonic analyses of yogurts were performed using a one-way ANOVA with Tukey’s test for assessing differences between samples. Calculations were conducted in the SPSS 25.0 statistical package (SPSS Inc., Chicago, IL, USA). Differences were considered significant at *p* ≤ 0.05.

## 3. Results and Discussion

### 3.1. Characterization of Wild Fruit Extracts: STE and HE

#### 3.1.1. Composition of Wild Fruit Extracts

The content of soluble proteins and reducing sugars in STE and HE is presented in Table 1. HE showed higher soluble proteins than STE (*p* < 0.05), although proteins represented minor components in both wild fruits. STE showed higher reducing sugar content (48.50 ± 5.10 g/100 g extract) than HE (33.24 ± 8.62 g/100 g extract) and showed significant differences (*p* < 0.05). Soluble sugars were the major components of both wild fruit extracts.

The presence of phenolic compounds in STE and HE, colorimetric methods (Folin assay) and HPLC combined with mass spectrometry (MS) were employed to determine the content of phenolic compounds.

The total phenol content assayed by the Folin method (Table 1) showed that values of polyphenols were in the same range for STE and HE (17.93 mg GAE/g and 22.01 mg GAE/g, *p* < 0.05, respectively). Although higher values were obtained in hawthorn, the results, however, were not statistically significant (*p* = 0.178). This phenolic content was in agreement with previous studies found in the literature. A study of different species of hawthorn fruits showed that the content of TPC in hawthorn fruits was significantly variable (*p* < 0.001) between species, ranging from 21.19 to 69.12 mg GAE/g dry weight. Fruits of *C. pentagyna* was highest value of total phenol content, while the lowest level was found in the fruits of *C. turkestanica* [41]. The TPC found for strawberry tree was similar to those previously reported [6,42,43]. Previous studies show a wide variation in total phenolic content among *A. unedo* genotypes grown in diverse agroclimatic conditions [6,44,45]. Different factors might influence the phenolic content, such as the class of plants (species, part used, and stage of development), technological processes (plant processing, concentration, time, and temperature of extraction), and environmental conditions (climate, season, and stresses), among others [46,47]. On the other hand, the literature describes different conditions of extraction of strawberry tree and hawthorn in an aqueous medium. For example, in the case of strawberry tree roots were boiled in water for 2 h [48], and roots and leaves were heated and boiled under reflux for 30 min [20]. On the other hand, in the case of hawthorn, conditions for preparing an aqueous extract at 80 °C for 20 min were previously described [49], as were conditions for preparing a water extract for 30 min at 70 °C by other authors [14]. We chose extraction at 60 °C for 1 h according to bibliographic references, and we also performed preliminary tests in our laboratory at different conditions of time and temperature (Appendix A).

The profile of phenolic compounds of STE and HE analyzed by HPLC-MS is shown in Table 2. A higher content of polyphenols was detected in HE. The phenolic profile between wild fruit extracts varied considerably, but some similarities were observed within families and/or genera. The main families found in our study in STE and HE extracts were flavonoids, phenolic acids, and anthocyanins. A higher content of flavonoids and anthocyanins was detected in HE, with significant differences (*p* < 0.05). The phenolic acid content was higher in STE extract (*p* < 0.05).

A total of 17 phenolic compounds were identified in STE. The phenolic profile of this extract agrees with previous studies that found catechin and cyanidin-3-glucoside in the same wild fruit collected. STE presented a higher content of phenolic acids, as the main compound detected was theogallin (1.89 ± 0.38 mg/g extract). However, other authors reported that gallic acid (10.7 mg/g DW) was the main phenolic compound in strawberry tree fruits [50,51]. Other phenolic acids previously identified in STE were protocatechuic acid, gentisic acid, p-hydroxybenzoic acid, vanillic acid, and m-anisic acid, and in minor quantities, p-hydroxybenzoic, vanillic, and m-anisic acids in the fruit [50]. A great variety of polyphenols were identified related to their mass spectrum and MS fragmentation pattern: flavanols (catechins, procyanidin dimers, and respective gallate esters), flavonols (glucosides of myricetin, quercetin, and kampherol), several galloyl (gallotannins), and ellagic (ellagitannins) derivatives [51]. Many of these compounds were already identified in *A. unedo* by other authors [6,50,52]. The anthocyanin composition of STE in our study was in the same range as reported in the literature, with cyanidin 3-galactoside being the most abundant compound [53].

A total of 16 phenolic compounds were identified in our HE samples. Our results showed that flavonoid compounds consisted primarily of epicatechin (8.12 ± 1.03 mg/g extract), followed by procyanidin dimer (4.34 ± 0.88 mg/g extract). The values obtained in our study for HE are higher than those described previously [9]. The authors observed that hawthorn is a source of phenolic compounds, with fractions of anthocyanins and phenolic acids as the major compounds. Procyanidins, flavanols, flavonols, C-glycosyl flavones, phenolic acids, anthocyanins, and lignans have been identified in different organs of hawthorn plants. In fruits, oligomeric procyanidins and their glycosides are the major phenolic compounds [54]. In this regard, procyanidins in hawthorn consist of primarily epicatechin as the flavan-3-ol unit, which is in agreement with our results [55]. In another study, hyperoside, chlorogenic acid, and isoquercetin were found to be the most abundant compounds present in hawthorn fruits [41]. However, Alirazalu *et al.* indicated that in most species, vitexin 2-O-rhamnoside was not detected and the quercetin content was very low, which is in concordance with our results [41].

Both in STE and in HE, cyanidin glucoside was the main anthocyanin detected, which was higher in HE (1.30 ± 0.13 mg/g extract) than in STE (0.11 ± 0.01 mg/g extract) (*p* < 0.05). The proportion of phenolic compound families was similar in both wild fruit extracts: flavonoids represented between 78 and 84% of the total phenolic compounds, phenolic acids between 11 and 17%, and anthocyanins between 1 and 6% for STE and HE, respectively. Only four phenolic compounds were detected in both STE and HE: catechin, procyanidin dimers I and III, and cyanidin glucoside. There were no common phenolic acid compounds identified. The phenolic compounds presence and concentration can be influenced by variation in fruit species and differences in growth conditions, genetic background, and methodological procedures [41]. On the other hand, due to the variety of analytical methods used and different extraction solvents used for the determination of the phenolic compounds, the comparison of the literature should be carefully conducted [56]. Moreover, the combination of both extracts could be a strategy to improve the phenolic compound profiles of both extracts obtained from these wild fruits.

The results obtained by the Folin–Ciocalteu method compared with HPLC showed some differences between wild fruit extracts. In the case of strawberry tree extracts, a higher content was detected by TPC (17.93 ± 4.17 mg GAE/g extract) than HPLC (4.74 ± 0.87 mg GAE/g extract). In the case of hawthorn, similar content was detected by the Folin–Ciocalteu method and HPLC (22.01 ± 1.16 mg GAE/g extract and 22.25 ± 2.77 mg/g extract, respectively). In general, the usual trend described in the literature was a higher content of total polyphenols determined by the Folin–Ciocalteu method than the content determined by HPLC [57,58]. The spectrophotometric method used for TPC, which is not totally specific for phenolic compounds, was only performed for the purpose of comparing between extracts; in order to accurately measure the amount of total phenolic compounds, chromatographic methods were used, which are more sensitive and specific than the spectrophotometric method. In this sense, the Folin–Ciocalteu method is the most widely used and is recognized as non-specific and differentially sensitive to different phenolic and flavonoid compounds. Furthermore, in plant extracts, other interfering compounds, such as sugars and ascorbic acid, would contribute to the total phenolic content [59]. However, they are not expected to affect the total phenols determined by HPLC. In addition, HPLC provides more specific information on individual compounds or groups. The high correlation between the HPLC and Folin–Cioacalteu methods could be important and useful in the estimation of phenols [60,61].

#### 3.1.2. Health Promoting Properties of Wild Fruit Extracts

Results regarding the antioxidant capacity, antidiabetic, and antihypertensive properties of the extracts are shown in Table 1. In this study, the antioxidant capacity of STE and HE was evaluated by ABTS and ORAC assays. HE extract showed higher antioxidant capacity measured by means of ABTS and ORAC methods. The antioxidant capacity cannot be fully reported by any one single method, so it is necessary to assay different antioxidant activity methods to take into account the various mechanisms of antioxidant action due to the fact that most natural antioxidants are multifunctional [62].

ABTS and ORAC assays are based on different chemical reactions: electron transfer reactions and proton transfer reactions, respectively. Overall, HE showed higher antioxidant capacity in both ABTS and ORAC assays, with significant differences by ORAC assay (*p* < 0.05). ABTS values of antioxidant capacity in STE and HE were higher than those previously reported in the literature for ethanolic strawberry tree extracts and methanol/water hawthorn extracts, respectively [9,63]. In our study, the correlation coefficient (r) between TPC and antioxidant capacity determined by ORAC was r = 0.98. This suggests that the antioxidant capacity found in the tested fruits was directly related to TPC, as fruits with higher TPC also presented higher values of antioxidant capacity. Namely, a high correlation coefficient was observed between flavonoids and anthocyanidin contents and the antioxidant capacity measured by the ORAC method (r = 0.98 and r = 0.88, respectively). These results are in agreement with the findings of many other authors who reported a positive correlation between TPC and antioxidant activity in different fruits and vegetables [8,9,63]. This suggests that the antioxidant activity in fruits is attributed to a greater extent to the type of individual phenolic compounds present than to the TPC. Our results also indicate that STE and HE are strong radical scavengers and can be considered potential sources of natural antioxidants. Therefore, the application of these extracts as food ingredients may have potential beneficial biological effects against chronic diseases related to oxidative stress, including diabetes mellitus, cancer, Alzheimer’s, atherosclerosis, etc. [64].

STE and HE were tested as potential digestive enzyme inhibitors. The inhibition of selected digestive enzymes by STE and HE is shown in Table 1. Numerous studies have shown a strong correlation between phenolic content, enzyme inhibition, and antioxidant properties [65]. Inhibition of enzymes such as α-amylase and α-glucosidase, which are implicated in carbohydrate digestion, can reduce the increase in blood post-prandial glucose, which is one of the most effective approaches for diabetes care. The most common synthetic inhibitor commercially available is acarbose. However, it is necessary to broaden the search for natural inhibitors of enzymes as an alternative to reduce and avoid unwanted secondary effects due to excessive enzymatic activity inhibition resulting in abnormal fermentation of undigested saccharides in the colon [65]. Moderate amylase inhibition coupled with strong glucosidase inhibition appears to be the ideal strategy to control the release of glucose from disaccharides in the intestine [30,66]. Therefore, to evaluate the hypoglycemic potential of *A. unedo* and *C. monogyna* wild berries, the extracts analyzed were assayed for their inhibitory effect against α-amylase and α-glucosidase activities. The inhibition of α-amylase by STE and HE at 10 mg/mL was very low, with values of inhibition under 10%. A higher inhibition was measured for STE extract (5.46 versus 6.02% for HE extract), although the differences were not statistically significant (*p* < 0.05). The inhibition of α-glucosidase by STE and HE was 54.07 and 57.92%, respectively. Significant differences (*p* < 0.05) between STE and HE were shown. Our results indicated that the extracts were more active towards α-glucosidase inhibition than α-amylase inhibition. Results showed that both wild fruits had low α-amylase inhibition (under 10%) and medium α-glucosidase inhibition (50–60%). The traditional use of strawberry tree as an antidiabetic agent may be associated with the presence of complex phytocompounds. A previous study indicated that wild fruit extracts were better at inhibiting α-amylase than other organs of the plant, like flowers [67]. Several studies suggested that the inhibitory effect against α-glucosidase may be associated with catechin, which is closely related to the presence of a free hydroxyl group at the 3-position of this molecule [68]. Catechin is a very common and widely diffused metabolite in the plant kingdom. In fact, *A. unedo* fruits are considered an alternative source of flavan-3-ols, in particular catechin and its derivatives [69]. However, their potential health-promoting properties are not due just to one molecule but to the presence of catechins and other phytoconstituents [70]. The traditional use of *A. unedo* roots as an antidiabetic agent was previously reported to have a potential α-glucosidase inhibitory activity greater than acarbose [70]. Different beneficial health-promoting effects of hawthorn extracts have been previously described in the literature [71,72,73]. The inhibitory activity of *Crataegus* species has previously been studied [74,75]. Miao et al. investigated different extracts prepared from hawthorn fruit using 80% ethanol, 80% methanol, 80% acetone, and pure deionized water. Their results indicate that extraction with a mixture of deionized water and 80% acetone preserved the activity of α-glucosidase better. These results supported the idea that deionized water extract has the potential to be employed as an ingredient in functional food products [76].

Several studies reported that phenolic acids such as caffeic acid, ferulic acid, syringic acid, ellagic acid, quercetin, pyrogallol, protocatechuic acid, vanillin, p-coumaric acid, and gallic acid present in fruits might be responsible for the inhibition of α-amylase and α-glucosidase activities and contribute to their antioxidant capacity. Numerous studies have reported the inhibition potential of α-amylase and α-glucosidase of phenolic acids [21,65,77,78]. Furthermore, the harvest time, the growing area, and the morphological part of the plant have an important effect on its phytochemical composition and biological activities such as antioxidant activity and antidiabetic capacity [79].

Lipase is an enzyme primarily produced in the pancreas that hydrolyzes lipids to form fatty acids in order to be absorbed by the digestive system. The inhibition of pancreatic lipase is the main prescribed treatment for weight management and obesity [80]. Our results showed that STE and HE inhibited the activity of the pancreatic lipase by over 90% in a dose-dependent manner. Results from previous studies have also observed the inhibitory activity of hawthorn against pancreatic lipase and pancreatic α-amylase *in vitro* [81]. Currently, the synthetic product Orlistat is clinically used as a pancreatic lipase inhibitor, although unwanted side effects such as flatulence, diarrhea, or dyspepsia are also commonly developed [82,83]. Therefore, it is necessary to search for alternative inhibitors from natural sources with minimal or no side effects [30,84,85]. Hawthorn has been employed as a treatment for digestive disorders, hyperlipidemia, high blood pressure, hypocholesterolemic, and lowering serum cholesterol [86]. In addition, hawthorn methanolic leaf extract compounds have lipid-lowering, antioxidant, anti-inflammatory, and protective effects against diabetes, as described in previous studies [87]. Also, some authors indicate that hawthorn could be useful for managing diabetes and obesity as it is a commonly available, inexpensive, and safe functional food [86]. Regarding strawberry tree fruits, we did not find previous results on their inhibitory effect against pancreatic lipase. To our knowledge, this is the first time that lipase inhibition has been evaluated in strawberry tree extracts. In this sense, more studies are necessary to analyze the effect *in vivo* of strawberry tree extracts on lipase activity and evaluate their activity against lipid digestion and absorption.

The antihypertensive activity of STE and HE had been studied *in vitro* using an Angiotensin Converting Enzyme (ACE) inhibitory assay. Angiotensin I-converting enzyme (ACE) is a peptidyl dipeptide hydrolase that plays an important physiological role in both the regulation of blood pressure and cardiovascular function. The results obtained for STE and HE are shown in Table 1. Our results showed that STE and HE inhibited the activity of the ACE by less than 25%. Hawthorn aqueous extracts from leaves and flowers have been reported to possess significant clinical effects in reducing blood pressure [88]. Different mechanisms were proposed to explain why hawthorn has antihypertensive activity; one of them indicated that flavonoids and proanthocyanidins present in hawthorn may have ACE inhibitory activity [89]. The suppression of ACE is considered a useful approach for the regulation of blood pressure. ACE inhibitors are extensively used as pharmaceutical drugs or components of functional foods for the treatment of cardiovascular diseases [90].

#### 3.1.3. Technological Properties of Wild Fruit Extracts

The technological properties of strawberry tree and hawthorn extracts were analyzed to evaluate their potential as food ingredients in the yogurt matrix. Some functional properties, mainly those related to protein content, are highly affected by pH changes. Both STE and HE had pH acid values (pH 3.85 for STE and pH 4.86 for HE). Similar values were found in the literature for strawberry tree [6,91] and higher in the case of hawthorn [41]. They studied the fruits of 15 samples of different hawthorn species (*Crataegus* spp.) collected from different regions of Iran. Their results demonstrated that the origin of the species had significant effects (*p* < 0.001) on the chemical characteristics of hawthorn fruits, including pH.

To evaluate the safety of STE and HE for their use as food ingredients, counts of total aerobic microorganisms, yeasts, and molds were measured. Counts lower than 4.5 log CFU/g were obtained in both extracts. Mold and yeast, enterobacteria, and coliform microorganisms were not detected (Appendix A).

### 3.2. Application of Wild Fruit Extracts as Food Ingredients in Yogurt

#### 3.2.1. Composition of Yogurts Containing Wild Fruit Extracts

The development of yogurts containing STE and HE was approached from a global perspective, taking into account the nutritional composition, technological, and hedonic properties of the products. We incorporated inulin-type fructans (inulin and fructo-oligosaccharides (FOS)) in yogurts to improve the texture and avoid adding sugar. The yogurt matrix was selected as it is consumed worldwide and can be incorporated without changing dietary patterns [16].

Results from the composition of yogurts containing STE and HE are shown in Table 3. The lactose content in yogurts ranged from 5.67 to 10.40 g/100 g. Yogurts with strawberry tree or hawthorn extract had higher protein content than yogurt controls (*p* < 0.05). No differences in reducing sugar content between samples were detected.

The TPC of the developed yogurts varied depending on the type of wild fruit extract that was used. The TPC of yogurts increased with the addition of wild fruit extracts although the difference was not statistically significant (*p* < 0.05) (Table 3). YHE-12 showed a higher TPC than strawberry tree yogurts, which may be due to the higher content of phenolic compounds in hawthorn extract. The total content of polyphenols in the analyzed samples is variable, and a dose-dependent effect was observed.

The phenolic profile previously identified in wild fruit extracts by HPLC-MS was maintained in yogurts containing STE and HE (Table 4). The main families identified in both wild fruit extracts were flavonoids, phenolic acids, and anthocyanins. Flavonoids were the predominant phenolic compound in yogurts containing HE, while in yogurts containing STE, phenolic acids were the main compounds present. Yogurts containing STE presented a higher content of phenolic acids, as the main compound detected was theogallin (13.01 ± 1.97 and 14.27 ± 0.99 µg/g yogurt, YSTE-8 and YSTE-12, respectively). Yogurts containing HE showed mainly flavonoids, primarily epicatechin (21.96 ± 0.72 and 35.35 ± 11.99 µg/g yogurt, YHE-8 and YHE-12, respectively), followed by procyanidin dimer (6.80 ± 0.41 and 9.89 ± 8.56 µg/g yogurt, YHE-8 and YHE-12, respectively). Yogurts containing HE presented higher levels of anthocyanins, mainly cyanidin glucoside, than YSTE. Anthocyanins are the primary pigment responsible for the reddish color of fig fruits [92]. Both wild strawberry tree and hawthorn showed a red color. It should be indicated that the intensity of the red color of both wild fruits could be related to the content of phenolic compounds. The addition of aqueous extracts of strawberry tree and hawthorn for the elaboration of different yogurts slightly affected the color, producing a certain pink color in the case of strawberry tree yogurts and a certain orange color in the case of hawthorn yogurts.

The phenolic content showed a dose-dependent linear effect, as yogurts containing higher concentrations of the extracts also showed higher TPC. However, the content of phytochemicals in yogurt in our study was lower than expected. This effect was also observed in yogurts containing polyphenol extracts from wine-making byproducts [21]. Other authors described yogurts with hawthorn and showed a range between 3.46 and 4.34 mg GAE/g yogurt. The TPC values are higher than our results [14]. The low observed phenolic content in yogurts could be due to several factors, such as the effect of the metabolic activity of yogurt starter culture bacteria on phenolic compounds, degradation of compounds, or matrix effects concerning protein–polyphenol interactions during milk fermentation. Some authors described four possible types of interactions that could happen between crude proteins and phenolic molecules: hydrogen bonding, ionic, hydrophobic, and covalent interactions. However, other factors may also affect protein–polyphenol interactions, such as molecular size, temperature of polyphenols, and pH [93]. Therefore, further analyses are needed to study the interactions that have been able to modify the phytochemical structures of the extracts in yogurt.

#### 3.2.2. Health Promoting Properties of Yogurts Containing Wild Fruit Extracts

The antioxidant, antidiabetic, and antihypertensive properties of yogurts containing STE and HE are shown in Table 3. The ultimate challenge in functional foods is the development of food products that provide health-promoting properties beyond basic nutrition without compromising their organoleptic properties. Yogurts containing wild fruit extracts showed a slight increase in their antioxidant capacity, which was significantly greater in yogurts containing HE when measured by ORAC (*p* < 0.005). Yogurts containing STE and HE had similar antioxidant capacities measured by ABTS. However, YHE-12 had greater antioxidant capacity measured by ORAC (*p* < 0.05) than the strawberry tree yogurts. Previous studies have also reported a decrease in the antioxidant capacity observed in yogurts compared to what is expected based on the antioxidant capacity measured in the extracts and the amount of extract added. The decrease in the observed antioxidant capacity was associated with a food matrix effect due to the association between milk proteins and polyphenols from wild fruit extracts [94]. High antioxidant capacity is noted in yogurts with the addition of hawthorn extracts due to the high content of phenolic compounds in the extract. The results of the antioxidant capacity test correlate with the total content of polyphenols. The highest correlation ratio was observed between flavonoids and antioxidant capacity in the ABTS (r = 0.78) and ORAC assays (r = 0.98). The correlations of phenolic acids to ABTS and ORAC were r = 0.74 and r = 0.43, respectively. Anthocyanins showed a higher correlation to the ORAC assay (r = 0.82) than the ABTS assay (r = 0.50). Since the radical scavenging efficiency of an antioxidant depends on its ability to form a stable radical itself, different sugar molecules may provide different molecular structures in which they may either enhance or diminish the stability and affect the potency. The antioxidant capacity of a compound depends upon which free radical or oxidant is used in the assay, and chemically different methods for measuring antioxidant activity will produce different hierarchies of antioxidants [95].

Due to the antioxidant capacity of strawberry tree fruits, their phenolic-rich extracts have been used as functional ingredients in processed meat products to protect them from oxidative degradation [8,96,97]. In addition, other applications, such as the enrichment of yogurts with fruit extract of *A. unedo*, have been reported, which improved the antioxidant activity and the survival of its microbial community without affecting the chemical and microbiological characteristics of yogurt [98].

On the other hand, antioxidants are one of the main protective mechanisms against the effects of free radicals in the body. Phenolic compounds are the largest group of antioxidant substances present in foods of plant origin. The intake of total phenols has been reported in different population groups; the results of the study showed that a consumption of phenolic compounds in the adult population ranging between 280 ± 130 and 2771 ± 1552 mg/day reflects a health benefit, while a consumption ranging from 322 ± 153 to 2861 mg/day exerts a benefit against different diseases [99]. In this context, the yogurts elaborated in our study with strawberry tree and hawthorn extracts contribute around 10% of the content of phenolic compounds necessary to exert some beneficial effect on health per day. In addition, the bioaccessibility of these compounds would have to be evaluated to determine the amount of bioactive compounds capable of reaching the intestine. In addition, there are other factors, such as the age of the consumers, that can affect the bioaccessibility and bioavailability of polyphenols.

Inhibition of α-amylase, α-glucosidase, and lipase activities by yogurt samples is shown in Table 3. To our knowledge, this is the first time that the *in vitro* inhibition of the enzymatic activity of digestive enzymes such as α-amylase, α-glucosidase, and lipase and antihypertensive enzymes such as ACE has been determined in yogurts containing wild fruit extracts (hawthorn and strawberry tree) with inulin and FOS. Previous studies have described the physicochemical, structural, and microbiological properties of yogurts containing STE and HE [14,98,100]. The inhibition of α-amylase was moderate and similar in all preparations of yogurt containing STE and HE, which ranged between 2.79 and 3.84% of inhibition. This enzymatic inhibition was significantly higher (*p* < 0.05) in yogurts containing strawberry tree and hawthorn extracts as compared to the control, although statistically significant differences were only detected between control yogurts and YSTE-12, YHE-8, and YHE-12. No differences were detected between wild-fruit yogurts at 8 or 12 mg/mL.

The inhibition of α-glucosidase activity was greater in yogurts containing strawberry tree and hawthorn extracts compared to the yogurt control, although only in YSTE-12 (*p* < 0.05). These results suggest that yogurts containing STE and HE could play a role in controlling the release of glucose from disaccharides in the gut [30,66]. The increase in inhibition of α-glucosidase activity in yogurts with extracts may be due to the phenolic compounds present in STE and HE, which showed a higher free radical-scavenging activity than the yogurt control.

Lipase inhibition values were also similar for all yogurt formulations (*p* > 0.05). The addition of STE and HE to yogurt did not result in a statistically significant increase in the enzymatic lipase inhibition (*p* > 0.05), although the mean of the lipase activity inhibition data of the samples that included extracts of strawberry tree and hawthorn was higher than that of the control, and also a dose-dependent effect was observed for the incorporation of the extracts in the yogurt. This could be attributed to molecular interactions between polyphenols and other components in the food matrix, such as proteins [21]. Similar results were obtained in yogurts supplemented with safflower petal extract (*Carthamus tinctorius* L.) [101]. Therefore, strawberry tree and hawthorn extracts supplemented with yogurt might alleviate hyperglycemia and hyperlipidemia by downregulating the digestion and absorption of carbohydrates and lipids. However, further analyses will be needed to better study the inhibition of the enzymatic activity of digestive enzymes in yogurts.

Similar percentages of ACE inhibition were shown in the control, strawberry tree, and hawthorn yogurts. No differences were detected between wild-fruit yogurts at 8 or 12 mg/mL (*p* > 0.05).

Different mechanisms have been proposed to explain the inhibition of enzyme activity by extracts of wild fruits. ACE enzymatic activities can be regulated by the binding of biomolecules such as polyphenols, flavonoids, and bioactive peptides onto the active binding sites of the enzyme [102]. The addition of some herbs to yogurts may have indirectly altered the hydrolysis of milk proteins by affecting the yogurt’s bacterial growth and metabolism. During fermentation, proteolysis occurs, the most important biochemical process that generates peptides that may have an ACE inhibitory effect. To support the potential uses of new herbal yogurts with anti-ACE activities together with conventional drug treatments, it is necessary to conduct more studies to characterize specific peptides generated during fermentation storage in the presence of herbs and to improve treatment for people with hypertension [103].

It would be necessary to carry out more studies to evaluate the bioavailability of the phenolic compounds present in yogurts made with extracts of wild fruits as well as their beneficial effects against chronic diseases in human intervention studies. The consumption of yogurts that incorporate strawberry tree and hawthorn extracts could be included as part of the diet; however, they could not be considered the only daily source of phenolic compounds. More studies are needed in this regard.

#### 3.2.3. Technological Properties of Yogurts Containing Wild Fruit Extracts

The characterization of pH, syneresis, and viscosity is shown in Table 3. The addition of wild fruit extracts did not significantly (*p* > 0.05) modify these parameters. Yogurt’s pH, moisture, syneresis, or viscosity values were similar for control and also YSTE-8, YSTE-12, YHE-8, and YHE-12 samples, which is in accordance with the previous literature [14,104]. The monitoring of pH in yogurts is very important in terms of product safety [19]. The addition of wild fruit aqueous extracts did not significantly affect the pH of the samples compared to the control yogurt. Syneresis is considered by many researchers to be one of the most important parameters indicating the quality of yogurt during storage. Our results show that the control yogurt samples had similar syneresis compared to yogurt samples with different levels of wild fruit aqueous extracts. One possible factor influencing these parameters is the ability of proteins to retain water and milk fat cells in the structure of yogurt [14].

Bacterial counts of the control and wild fruit extract yogurts were higher than the minimum of 7 log CFU/mL legally required in yogurt manufacture by the Codex Alimentarius. The initial *S. thermophilus* counts (7.76–7.94 log CFU/mL) were significantly higher (*p* < 0.001) than those obtained for *L. delbrueckii* subsp. *bulgaricus* (5.54–6.07 log CFU/mL, respectively). To minimize the acetic acid taste produced by the metabolism of *L. delbrueckii* subsp. *bulgaricus,* lower counts of *L. delbrueckii* subsp. *bulgaricus* than *S. thermophilus* are common in yogurt elaboration with commercial cultures of bacteria [105]. The *S. thermophilus* counts in YSTE-8 and YSTE-12 were similar (7.81 and 7.86 log CFU/mL, respectively) to those obtained in YHE-8 and YHE-12 (7.94 and 7.76 log UFC/mL). The initial *L. delbrueckii* subsp. *bulgaricus* counts in YSTE-8 and YSTE-12 were similar (6.07 and 5.89 log CFU/mL, respectively) to those obtained in YHE-8 and YHE-12 (5.79 and 5.91 log UFC/mL).

The instrumental texture parameters of functional yogurts are shown in Figure 1. No statistically significant differences (*p* > 0.05) were observed between control and wild-fruit extract yogurts. However, a slight tendency to decrease the texture parameter values of yogurts containing dietary fiber when fruit extracts were included was observed. However, no effect of fruit concentration on texture parameters was observed. High firmness and consistency values have been found for yogurts similar to Greek yogurt, probably due to the texturizing properties of inulin [106,107].

#### 3.2.4. Hedonic and Sensory Analyses of Yogurts Containing Wild Fruit Extracts

Results from the hedonic analyses with untrained consumers (*n* = 104) are shown in Figure 2. Results showed that all yogurt formulations obtained similar scores (*p* > 0.05) in terms of smell (Figure 2a), taste (Figure 2b), and texture (Figure 2c). No statistically significant differences were detected between the yogurts for smell (*p* ≤ 0.05). In the case of taste evaluation, the control yogurt obtained higher scores than the yogurts with HE extracts. No statistically significant differences were detected for taste evaluation between the yogurt control and YSTE-8 and YSTE-12. In the case of texture evaluation, the control yogurt obtained a higher score than the yogurts with HE extracts, although no statistically significant differences were detected between the yogurt control and YSTE-8 and YSTE-12. The overall liking of yogurt formulations (Figure 2d) obtained a value over 6, which suggests that yogurts were well accepted by consumers. Several authors indicated high overall linking in hedonic analyses when they obtained scores over 6 in different food products [21,108,109]. The control yogurt only obtained a significantly higher overall liking score (*p* < 0.05) compared to the yogurts containing HE. This high acceptance of yogurt control could be due to familiarity with conventional yogurt (Figure 2d). Furthermore, the incorporation of wild fruit extracts in yogurts may affect different sensory attributes because they are not very common flavors. Sugar and fruity or floral aromas interact with each other in the form of potentiation. In these senses, as described in the literature, a large amount of sugar needs to be added for flavoring when producing hawthorn products due to the excessive organic acid content in hawthorn [47]. Also, it was previously described that there was a differential effect of tagatose and sucrose on flavor perception despite the equal level of sweetness in strawberry yogurt [110]. In the yogurt samples in the present study, the incorporation of ingredients like inulin and FOS may have potentially enhanced the wild fruit flavor at the time of tasting. Further investigations on the effects of wild fruit extracts on flavor release are needed to study these effects.

Although the control yogurts were the most acceptable products in our study, if we compare only the different yogurts made with wild fruit extracts, YSTE-8 showed higher acceptability in flavor, texture, and overall acceptability than YHE-12, which obtained the worst acceptance for these parameters. The results seem to indicate that the effect of concentration in the preparation of the different yogurts affects the acceptability of the final product. The lower added concentrations of strawberry tree extract and hawthorn extract in the yogurts produced (YSTE-8 and YHE-8, respectively) were better accepted in terms of texture and overall acceptability than yogurts produced with higher concentrations of both types of extracts (YSTE-12 and YHE-12). In the case of flavor, YSTE-8 was more acceptable than YSTE-12, showing significant differences. There were no significant differences in any of the parameters studied in the yogurts from hawthorn elaborated at different concentrations (YHE-8 and YHE-12). In the literature, differences were described in terms of acceptance about the taste, smell, and appearance of yogurts fortified with plant additives. The final evaluation of the finished product depends on the external appearance of the additive itself and the quantity in which it was used. Available reports indicate that enriching yogurts with herbs or green parts of plants significantly deteriorates their taste. Reduced palatability, consistency, and appearance negatively impact the overall sensory evaluation. In general, the sensory evaluation of the product tends to deteriorate with the increased presence of additives. It is therefore important to conduct test production to optimize the amount of the additive when designing a new product [111].

It is important to note that nutritional, safety (microbiological and toxicological), sensory, and commercial (price) considerations should be taken into consideration when dairy foods with herbal extracts are manufactured [93]. We proposed the application of two wild fruit extracts as ingredients in yogurts. To elaborate on sugar-free dairy products, we used inulin and FOS to improve sensory properties of yogurts. Inulin improves yogurt texture and mouthfeel, and FOS are more soluble and sweeter than inulin. The combination of these compounds was previously described and well accepted by consumers [21].

## 4. Conclusions

This study showed that strawberry tree and hawthorn extracts have interesting properties (antioxidant capacity, inhibition of metabolic enzymes, and phenolic compound content) suitable for their use as food ingredients.

These extracts could be valuable potential sources of safe antioxidants of natural origin. Therefore, it seems important to provide information on the bioactive compound contents and their antioxidant capacity of these fruits to help them recover and increase their regular consumption.

The extracts of strawberry tree and hawthorn can be employed together with inulin and FOS in the development of yogurt as sources of bioactive compounds and dietary fiber. Moreover, the new yogurts would allow for the “high in fiber” nutritional claim due to the inulin and FOS as ingredients.

The incorporation of strawberry tree or hawthorn extracts as ingredients for the manufacture of yogurt increased the antioxidant capacity and the inhibition of alpha-glucosidase enzymatic activity of the yogurts. The hawthorn yogurt presented the highest overall antioxidant capacity, and yogurts with hawthorn and strawberry tree extracts, both at 8 mg/mL and at 12 mg/mL, showed the same inhibition of alpha-glucosidase activity.

All yogurt formulations were considered acceptable by consumers and obtained over 6 scores in smell, texture, and overall acceptance; however, the taste acceptability of the hawthorn yogurts scored less than 6 points. The control yogurt received higher overall acceptance scores than the yogurts that included hawthorn or strawberry tree extracts. The results obtained from the sensory analyses of yogurts elaborated with wild fruit extracts showed the YSTE-8 to be the most accepted product. Therefore, the results of this study showed that STE and HE may be combined with inulin and FOS for the development of health-promoting yogurts.

## Figures and Tables

**Figure 1 foods-12-03332-f001:**
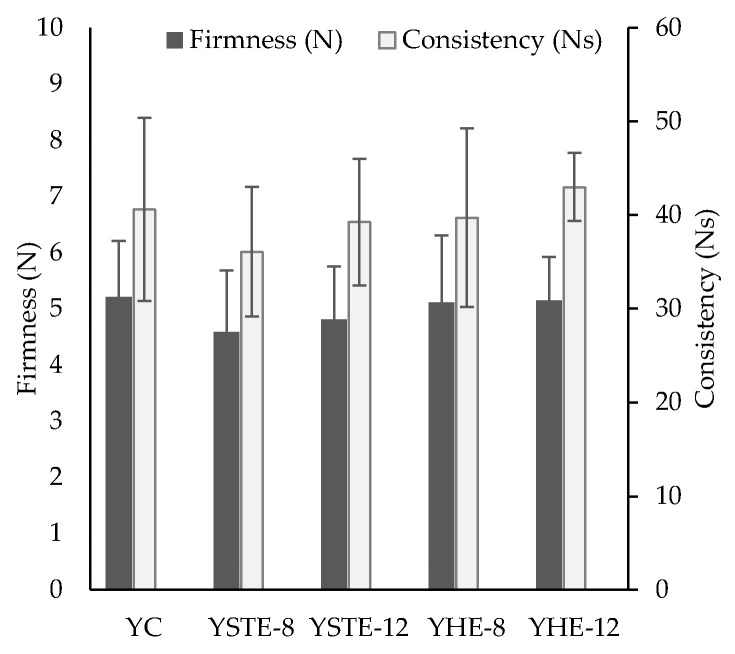
Instrumental firmness (N) and consistency (Ns) of a control yogurt (C), yogurt with strawberry tree extract (YSTE-8 and YSTE-12), and yogurt with hawthorn extract (YHE-8 and YHE-12). Different letters denote significant differences (*p* < 0.05).

**Figure 2 foods-12-03332-f002:**
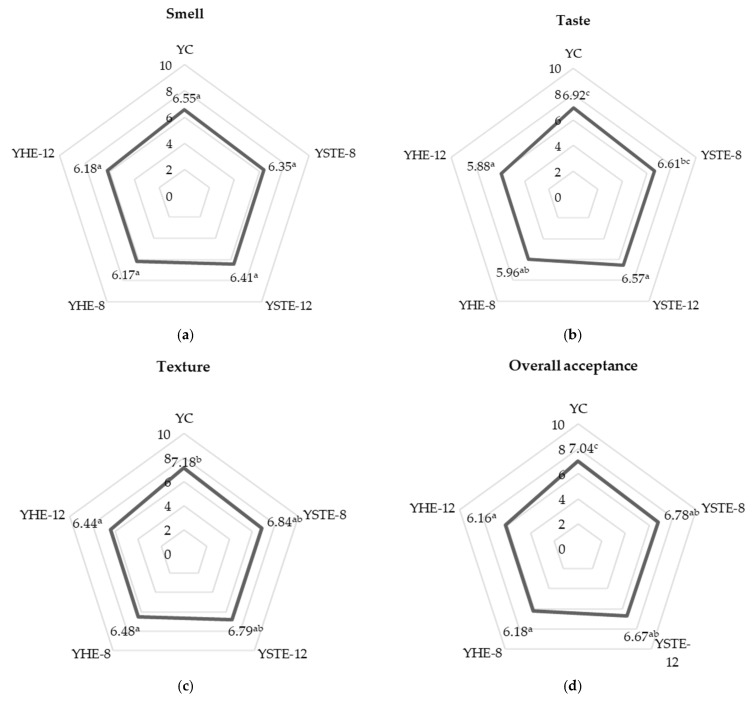
Spider-web diagram, which shows mean scores (*n* = 104) of a 1–10 scale sensorial analysis of yogurts: (**a**) Smell; (**b**) Taste; (**c**) Texture; (**d**) Overall acceptance. Different letters denote significant differences (*p* < 0.05).

**Table 1 foods-12-03332-t001:** Nutritional composition: proteins and reducing sugars (g/100 g extract), total phenolic content (TPC) (mg GAE/g extract), antioxidant capacity (µmol TE/g extract), and inhibitory digestive enzymes of strawberry tree extracts (STE) and hawthorn extracts (HE).

Chemical	STE	HE
Proteins (g/100 g extract)	1.40 ± 0.06 ^a^	2.62 ± 0.10 ^b^
Reducing sugar (g/100 g extract)	48.50 ± 5.10 ^b^	33.24 ± 8.62 ^a^
TPC (mg GAE/g extract)	17.93 ± 4.17 ^a^	22.01 ± 1.16 ^a^
Antioxidant capacity		
ABTS (µmol TE/g extract)	573.24 ± 10.31 ^a^	731.00 ± 32.29 ^a^
ORAC (µmol TE/g extract)	190.39 ± 7.35 ^a^	625.94 ± 27.60 ^b^
Antidiabetic properties		
α-Amylase inhibition (%)	5.46 ± 0.65 ^a^	6.02 ± 0.44 ^a^
α-Glucosidase inhibition (%)	54.07 ± 9.90 ^a^	57.92 ± 0.92 ^b^
α-Glucosidase IC50 (mg/mL)	7.26 ± 0.34 ^a^	8.01 ± 0.27 ^b^
Lipase inhibition (%)	97.79 ± 15.74 ^a^	91.01 ± 2.88 ^a^
Lipase IC50 (mg/mL)	8.14 ± 0.50 ^b^	3.63 ± 0.37 ^a^
Antihypertensive properties		
ECA inhibition (%)	20.71 ± 1.83 ^b^	14.27 ± 1.03 ^a^

Results are reported as mean ± SD (*n* = 3). Different letters within the same row denote statistically significant differences (*p* ≤ 0.05).

**Table 2 foods-12-03332-t002:** Phenolic profiles of STE and HE identified and quantified by HPLC-MS (mg/g extract).

Compounds	TR	M/Z [M-H]	Formula	STE	HE
Catechin	9.8	289.0718	C_15_H_14_O_6_	0.40 ± 0.03 ^a^	0.19 ± 0.03 ^b^
Epicatechin	14.4	289.0718	C_15_H_14_O_6_	0.01 ± 0.00 ^a^	8.12 ± 1.03 ^b^
Quercetin Galactoside	23.7	463.0882	C_21_H_20_O_12_	n.d.	2.22 ± 0.34
Quercetin Glucoside	24.4	463.0882	C_21_H_20_O_12_	n.d.	1.36 ± 0.19
Quercetin Glucoside Acetate	25.8	505.0988	C_23_H_22_O_13_	n.d.	0.14 ± 0.02
Procyanidin Dimer (I)	8.2	577.1351	C_30_H_26_O_12_	0.10 ± 0.01	n.d.
Procyanidin Dimer (II)	8.9	577.1351	C_30_H_26_O_12_	0.07 ± 0.01	n.d.
Procyanidin Dimer (III)	12.7	577.1351	C_30_H_26_O_12_	n.d.	4.34 ± 0.88
Procyanidin Dimer (IV)	14.3	577.1351	C_30_H_26_O_12_	0.01 ± 0.00	n.d.
Procyanidin Dimer (V)	17.9	577.1351	C_30_H_26_O_12_	0.01 ± 0.00	n.d.
Quercetin Rutinoside	23.2	609.1461	C_27_H_30_O_16_	n.d.	0.25 ± 0.88
Quercetin Diglucoside	19	625.141	C_27_H_30_O_17_	n.d.	0.09 ± 0.01
Procyanidin Trimer	16.81	865.1985	C_45_H_38_O_18_	n.d.	1.52 ± 0.38
TOTAL Flavonoids				0.59 ± 0.06 ^a^	18.38 ± 2.37 ^b^
Fertaric Acid (I)	4.6	325.0565	C_14_H_14_O_9_	0.07 ± 0.01	n.d.
Hydroxybenzyl Tartaric Acid	4.8	255.051	C_11_H_12_O_7_	n.d.	1.15 ± 0.16
Fertaric Acid (II)	5.1	325.0565	C_14_H_14_O_9_	0.25 ± 0.01	n.d.
Galloyl Glucose	4.3	331.0671	C_13_H_16_O_10_	0.40 ± 0.10	n.d.
Coumaroylquinic Acid	9.1	337.0929	C_16_H_18_O_8_	n.d.	0.22 ± 0.02
Theogallin	2.8	343.0671	C_14_H_16_O_10_	1.89 ± 0.38	n.d.
Caffeoylquinic Acid (I)	6.4	353.0887	C_16_H_18_O_9_	n.d.	0.32 ± 0.06
Caffeoylquinic Acid (II)	10.3	353.0887	C_16_H_18_O_9_	n.d.	0.70 ± 0.12
Caffeoylquinic Acid (III)	11.1	353.0887	C_16_H_18_O_9_	n.d.	0.12 ± 0.02
Caffeoylquinic Acid (IV)	13.9	353.0887	C_16_H_18_O_9_	n.d.	0.05 ± 0.00
Ellagic Acid Arabinoside	22	433.0412	C_19_H_14_O_12_	0.13 ± 0.05	n.d.
Ellagic Acid Glucoside	17.1	463.0518	C_20_H_16_O_13_	0.11 ± 0.04	n.d.
DigalloylShikimic Acid	12.9	477.0675	C_21_H_18_O_13_	0.43 ± 0.21	n.d.
Theogallin Derivative	9.2	495.0780	C_21_H_20_O_14_	0.69 ± 0.37	n.d.
Strictinin	21.2	581.2240	C_28_H_38_O_13_	0.06 ± 0.01	n.d.
TOTAL Phenolics acids				3.98 ± 0.82 ^b^	2.57 ± 0.27 ^a^
Cyanidin Glucoside	11.6	447.0933	C_21_H_21_O_11_	0.11 ± 0.01 ^a^	1.30 ± 0.13 ^b^
Cyanidin Arabinoside	13.7	419.0973	C_20_H_19_O_10_	0.06 ± 0.00	n.d.
TOTAL Anthocyanins				0.17 ± 0.01 ^a^	1.30 ± 0.13 ^b^
Total Phenolic Compounds				4.74 ± 0.87 ^a^	22.25 ± 2.77 ^b^

Results are reported as mean ± SD (*n* = 3). Different letters within the same row denote statistically significant differences (*p*   ≤  0.05). n.d.—no detected.

**Table 3 foods-12-03332-t003:** Nutritional composition: proteins, reducing sugars, and lactose (%), total phenolic content (TPC) (mg GAE/g extract), antioxidant capacity (mg TE/g extract), inhibitory activity of digestive enzymes (%), physicochemical parameters, and apparent viscosity of strawberry tree yogurts (YSTE) and hawthorn yogurts (YHE).

		Strawberry Tree Yogurts		Hawthorn Yogurts	
	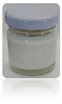 **Control**	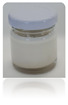 **YSTE-8**	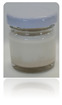 **YSTE-12**	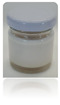 **YHE-8**	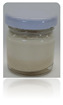 **YHE-12**
Proteins (%)	1.49 ± 0.01 ^a^	2.10 ± 0.04 ^b^	2.07 ± 0.19 ^b^	1.99 ± 0.07 ^b^	2.10 ± 0.01 ^b^
Reducing sugar (%)	3.36 ± 0.04 ^a^	3.50 ± 0.15 ^a^	3.55 ± 0.26 ^a^	3.11 ± 0.00 ^a^	3.25 ± 0.00 ^a^
Lactose (%)	9.67 ± 0.42 ^a^	10.13 ± 0.70 ^a^	10.40 ± 0.35 ^a^	5.67 ± 0.3 ^a^	6.13 ± 0.31 ^a^
TPC (mg GAE/g yogurt)	0.07 ± 0.02 ^a^	0.11 ± 0.04 ^a^	0.10 ± 0.04 ^a^	0.12 ± 0.05 ^a^	0.17 ± 0.09 ^a^
Antioxidant capacity					
ABTS (µmol TE/g extract)	0.59 ± 0.25 ^a^	0.85 ± 0.22 ^a^	0.89 ± 0.27 ^a^	0.87 ± 0.24 ^a^	0.96 ± 0.26 ^a^
ORAC (µmol TE/g extract)	0.58 ± 0.09 ^a^	2.98 ± 0.36 ^a^	2.69 ± 0.32 ^a^	8.61 ± 0.70 ^b^	15.82 ± 2.56 ^c^
Antidiabetic properties					
α-Amylase inhibition (%)	2.79 ± 0.23 ^a^	3.10 ± 0.48 ^ab^	3.53 ± 0.45 ^b^	3.29 ± 0.15 ^b^	3.84 ± 0.07 ^b^
α-Glucosidase inhibition (%)	9.96 ± 0.86 ^a^	23.84 ± 6.34 ^ab^	27.55 ± 3.88 ^b^	22.69 ± 5.09 ^ab^	23.09 ± 4.92 ^ab^
Lipase inhibition (%)	44.14 ± 6.44 ^a^	49.48 ± 2.39 ^a^	50.86 ± 4.00 ^a^	45.38 ± 2.22 ^a^	46.17 ± 5.35 ^a^
Antihypertensive properties					
ACE inhibition (%)	16.37 ± 2.67 ^a^	15.65 ± 1.40 ^a^	15.69 ± 0.80 ^a^	16.50 ± 2.79 ^a^	17.49 ± 1.24 ^a^
Physicochemical parameters					
pH	4.50 ± 0.06 ^a^	4.49 ± 0.09 ^a^	4.50 ± 0.02 ^a^	4.57 ± 0.04 ^a^	4.51 ± 0.07 ^a^
Moisture (%)	83.60 ± 1.97 ^a^	84.04 ± 1.55 ^a^	83.04 ± 0.51 ^a^	83.81 ± 1.12 ^a^	83.62 ± 0.40 ^a^
Syneresis (%)	50.67 ± 3.92 ^a^	48.77 ± 7.99 ^a^	53.19 ± 4.94 ^a^	55.54 ± 0.84 ^a^	51.98 ± 2.69 ^a^
Viscosity (Pas)	3.70 ± 0.39 ^a^	3.53 ± 0.89 ^a^	3.35 ± 0.96 ^a^	3.01 ± 0.26 ^a^	3.40 ± 0.72 ^a^

Results are reported as mean ± SD (*n* = 3). Different letters within the same row denote statistically significant differences (*p*  ≤  0.05).

**Table 4 foods-12-03332-t004:** Phenolic profile of strawberry tree (YSTE) and hawthorn (YHE) yogurt (µg/g yogurt).

	Compounds	Strawberry Tree Yogurts		Hawthorn Yogurts	
		YSTE-8	YSTE-12	YHE-8	YHE-12
Flavonoids	Catechin	0.63 ± 0.17 ^b^	0.73 ± 0.08 ^b^	0.16 ± 0.02 ^a^	0.16 ± 0.02 ^a^
	Epicatechin	0.03 ± 0.00 ^a^	0.03 ± 0.01 ^a^	21.96 ± 0.72 ^b^	35.35 ± 11.99 ^b^
	Quercetin Galactoside	n.d.	n.d.	1.70 ± 0.27	3.29 ± 0.59
	Quercetin Glucoside	n.d.	n.d.	0.82 ± 0.03	1.76 ± 0.69
	Quercetin Glucoside Acetate	n.d.	n.d.	0.02 ± 0.00	0.04 ± 0.00
	Procyanidin Dimer (I)	0.05 ± 0.02	0.06 ± 0.00	n.d.	n.d.
	Procyanidin Dimer (II)	0.05 ± 0.01	0.08 ± 0.00	n.d.	n.d.
	Procyanidin Dimer (II)	n.d.	n.d.	6.80 ± 0.41	9.89 ± 8.56
	Procyanidin Dimer (III)	0.00 ± 0.00 ^a^	0.00 ± 0.00 ^a^	0.00 ± 0.00 ^a^	0.00 ± 0.00 ^a^
	Procyanidin Dimer (IV)	0.00 ± 0.00 ^a^	0.00 ± 0.00 ^a^	n.d.	n.d.
	Quercetin Rutinoside	n.d.	n.d.	0.33 ± 0.01	0.56 ± 0.13
	Quercetin Diglucoside	n.d.	n.d.	0.00 ± 0.00	0.00 ± 0.00
	Procyanidin Trimer	n.d.	n.d.	0.15 ± 0.04	0.33 ± 0.01
TOTALFlavonoids		0.76 ± 0.21 ^a^	0.91 ± 0.10 ^a^	32.09 ± 1.47 ^b^	51.47 ± 22.58 ^b^
Phenolics acids	Hydroxybenzyl Tartaric Acid	n.d.	n.d.	3.68 ± 0.07	8.81 ± 0.24
	Fertaric Acid (I)	0.34 ± 0.07	0.36 ± 0.06	n.d.	n.d.
	Fertaric Acid (II)	1.44 ± 0.31	1.57 ± 0.21	n.d.	n.d.
	Galloyl Glucose	1.38 ± 0.30	1.56 ± 0.20	n.d.	n.d.
	Coumaroylquinic Acid	n.d.	n.d.	0.85 ± 0.00	1.76 ± 0.06
	Theogallin	13.01 ± 1.97 ^a^	14.27 ± 0.99 ^a^	n.d.	n.d.
	Caffeoylquinic Acid (I)	n.d.	n.d.	1.82 ± 0.07	3.57 ± 0.76
	Caffeoylquinic Acid (II)	n.d.	n.d.	2.02 ± 0.04	3.57 ± 0.83
	Caffeoylquinic Acid (III)	n.d.	n.d.	0.64 ± 0.00	19.69 ± 0.28
	Caffeoylquinic Acid (IV)	n.d.	n.d.	0.41 ± 0.06	0.93 ± 0.16
	Ellagic Acid Arabinoside	0.05 ± 0.01	0.05 ± 0.02	n.d.	n.d.
	Ellagic Acid Glucoside	0.07 ± 0.01	0.08 ± 0.03	n.d.	n.d.
	DigalloylShikimic Acid	0.28 ± 0.14	0.29 ± 0.10	n.d.	n.d.
	Theogallin Derivative	0.63 ± 0.24	0.70 ± 0.24	n.d.	n.d.
	Strictinin	0.12 ± 0.03	0.19 ± 0.01	n.d.	n.d.
TOTALPhenolic sacids		17.31 ± 3.06 ^ab^	19.07 ± 1.82 ^ab^	9.43 ± 0.02 ^a^	25.91 ± 2.33 ^c^
Anthocyanins	Cyanidin Glucoside	0.27 ± 0.06 ^a^	0.34 ± 0.04 ^a^	3.81 ± 2.20 ^b^	3.33 ± 2.41 ^b^
	Cyanidin Arabinoside	0.06 ± 0.01	0.07 ± 0.01	n.d.	n.d.
TOTALAnthocyanin		0.33 ± 0.07 ^a^	0.41 ± 0.05 ^a^	3.81 ± 0.09 ^b^	3.33 ± 1.41 ^b^
Total Phenolic Compounds		18.40 ± 3.28 ^a^	20.39 ± 1.96 ^a^	45.33 ± 21.78 ^b^	93.13 ± 26.99 ^b^

Results are reported as mean ± SD (*n* = 3). Different letters within the same row denote statistically significant differences (*p*  ≤  0.05). n.d.—not detected.

## Data Availability

Data is contained within the article.

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
