# Peer review of "Effect of Wild Strawberry Tree and Hawthorn Extracts Fortification on Functional, Physicochemical, Microbiological, and Sensory Properties of Yogurt"

_foods, 2023, doi:10.3390/foods12183332_

Round 1

Reviewer 1 Report

Comments and Suggestions for Authors

The submitted article reported the development and characterization of aqueous extracts of the strawberry tree and hawthorn for their introduction in developing polyphenol-enriched yogurts. With the aim t to characterize chemical composition, microbiological analysis, antioxidant properties and inhibitory effect against digestive enzymes of wild fruits extracts, chemical composition, microbiological analysis, antioxidant properties and inhibitory effect against digestive enzymes of yogurts with selected wild fruits extracts, and to evaluate the consumer sensory acceptance and purchase intent of these yogurts elaborated with wild fruits extracts and also dietary fiber. The topic is interesting for the selected journal, and the article is written correctly.

The introduction section provides enough background information relevant to the article's topic.

The methods are given in detail.

The result and discussion sections are comprehensive with proper comparison with the results from the literature.

The conclusion section gives the essence of the presented research.

There are more than enough references.

Specific comments:

Page 2, line 93, add the water content of the fresh fruits.

Page 2, line 95, provide a detailed description of the lyophilization process, state the used device, and add processing conditions.

Page 3, lines 97-98, explain why choosing extraction at 60C for 1 hour, and add a reference.

Page 3, lines 105-109, explain the selected concentrations for adding the wild fruit extract to the yogurts. Were they selected according to the literature or based on the preliminary research?

Did adding aqueous extracts of strawberry tree and hawthorn to yogurt affect the final product's color? Please, elaborate.

Reviewer 2 Report

Comments and Suggestions for Authors

The submitted manuscript on “Effect of wild strawberry and hawthorn extracts fortification on functional, physicochemical, microbiological and sensory properties of yogurt” is a relevant work on the development of bioactive products. The basic products and yoghurts were intensively investigated, and many aspects considered. Nevertheless, there are some weaknesses that have to be addressed and the manuscript should be thoroughly revised:

 L25-26 Why is it especially in the Mediterranean region and which cultures – explain. The first sentence sounds incomplete and general.

L27-30 compared to what? This is again too unspecific and general – what exactly is meant and give some examples.

L32-39 For Strawberry-tree there are more recent sources e.g.: doi: 10.3390/foods11233838

L37-39 sentences sounds a bit odd – should be rerwritten

Methods why a higher temperature of 60°C for extracts was used– for many compounds this is not suitable (e.g. ascorbic acid), compare with: https://doi.org/10.1016/j.foodres.2018.04.061, a reference or an explanation should be provided for the used method

L105-109 How visually different were the yoghurts? This might also affect the hedonic response of consumers and pictures should be added to the manuscript.

Spaces should be used in a similar way between numbers and units and references e.g. 

g glucose/ 100 g  vs. 4 g/100 mL AND microplate[25]. vs. analyzed [26–28].

L145 a ) is missing

L227 is there more information on the age and gender of the participants and was a consent form obtained from all participants? Revise this issue here also according to the author guidelines of Foods

L238 statistical analysis the analysis of the hedonic testing should be added.

L250 p-value missing for reducing sugar content

L268 the extraction temperature should be discussed a bit more – are 60°C already to destructive for some phenolic compounds,

Was it hypothesised as well that the combination of both compounds might result in more complete phenolic compound spectra according to Table 2? If not, you can consider it as the compounds seems to complement each other.

Why is the TPC of STE almost as high as of HE but the total phenolic compounds such lower. Are the phenolic compounds missing? Is there another explanation? This should be discussed.

L476 identifiedin – space is missing

L628 reference needed here, also for the 10cm scale, a 9-point hedonic scale is more common, explanation is needed why the other was used? Also, a reference is needed for the statement (on a 10cm hedonic scale) that a value over 6 is considered as well accepted.

L630 or could the lower acceptance of the flavored yogurts also by the ingredients? Was the taste described by a panel? Hawthorn and strawberry tree is not very common flavor for tasters? This should be discussed.

L624 from the figure 2c a significant difference is mentioned with different letters. This should be clarified and discussed.

Overall, it seems that the YSTE-8 was the most accepted product of the flavored ones and the YHE-12 the least accepted one? Do you agree? Interpret the sensory results more as they might help to increase a possible purchase behaviour

References and further discussion are needed of the tasting results.

According to results of the tasting the conclusion should be rewritten regarding the comment for L628

L660 Reference is required

Purchase intention was not presented in the results but mentioned in the conclusion and the methods section (as hypothesis) this should be considered in results and discussion section

Furthermore, the discussion section should include discussion on the health benefit of the actual used (final) concentration of the wild fruit extracts in the yoghurt (what do you expect? How many yoghurts per day for how long?) and also an explanation for the use of these concentrations should be provided (L105-109) incl. a reference

Round 2

Reviewer 2 Report

Comments and Suggestions for Authors

Dear authors,

The manuscript has been thoroughly revised and the quality has been sufficiently improved. 

best regards
